# LLM-MEDIATED PATHOLOGY MODELS FOR ROBUST CROSS-INSTITUTION GENERALIZATION

## ABSTRACT

Pathology foundation models (PFMs) have shown strong potential across clinical and scientific applications. Their performance, however, is often limited by batch effects, which are non-biological variations across tissue source institutions (TSIs) that distort feature representations and reduce generalization. Existing mitigation methods, such as stain normalization, have limited success in addressing these high-dimensional and complex artifacts. We introduce the General-purpose LLM-Mediated Pathology model (GLMP), a novel framework that generates robust numerical embeddings from histology image patches by first converting them into text descriptions. By leveraging pretrained multimodal large language models (MLLMs) and text encoders, GLMP prioritizes genuine biological signals over TSI-specific signatures and improves cross-TSI generalization compared to existing PFMs. Our findings demonstrate the effectiveness of broad-domain, non-specialized MLLMs in computational pathology and provide an alternative framework for developing versatile, generalizable, and robust pathology models that do not require large-scale, histology-specific pretraining data. Code is provided in Supplementary Materials for reproducibility and will be released to the public upon paper acceptance.

## 1 INTRODUCTION

Pathology foundation models (PFMs), pre-trained on large-scale unannotated histology data, have reshaped computational pathology by enabling diverse downstream clinical and scientific tasks. State-of-the-art PFMs are primarily vision encoders trained via self-supervised learning on hematoxylin-and-eosin (H&E) whole-slide images (WSIs) (Chen et al., 2024; Xu et al., 2024a; Nechaev et al., 2024; Zimmermann et al., 2024). Multimodal extensions integrate additional stains (Dippel et al., 2024; Hua et al., 2024), molecular profiles (Xu et al., 2024b; Vaidya et al., 2025), or paired image-text data (Lu et al., 2024b; Xiang et al., 2025). Pairing these encoders with large language models (LLMs) powers generative systems such as PathChat (Lu et al., 2024a), SlideChat (Chen et al., 2025), and PRISM2 (Shaikovski et al., 2025). Collectively, these capabilities position PFMs as a transformative tool for applications such as tissue classification, disease subtyping, and survival analysis, with promising potential for supporting clinical decision-making and advancing precision medicine in pathology.

Yet PFMs are susceptible to batch effects, which are non-biological artifacts linked to tissue source institutions (TSIs) (Howard et al., 2021; Kömen et al., 2024; de Jong et al., 2025) that commonly arise from variations in tissue fixation, processing, staining, and scanner hardware (Hoque et al., 2024; Chai et al., 2025). These artifacts can correlate with disease severity or other clinical variables, making TSI-specific signatures difficult to disentangle from biological signals. Consequently, state-of-the-art PFMs encode TSI information more strongly than cancer status (de Jong et al., 2025), which can lead to overestimation of accuracy by exploiting site-specific shortcuts (Howard et al., 2021) and propagating clinical bias when institutional signatures are associated with clinical variables of interest (Dehkharghanian et al., 2023). Mitigating batch effects within the embedding space is therefore critical for generalizable and safe deployment.

Various techniques have been developed to mitigate batch effects in histology images. These solutions primarily rely on color normalization (Reinhard et al., 2001; Macenko et al., 2009; Vahadane et al., 2016) and augmentation strategies that broaden model exposure to stain variability (Shen

et al., 2022; Marini et al., 2023). However, as these pixel-level remedies cannot remove higher-order artifacts introduced by tissue processing and scanner hardware, TSI signatures persist in learned representations of most PFMs (Keller et al., 2023; Kömen et al., 2024).

To overcome the limitations of existing PFMs and batch effect correction methods, we developed the General-purpose LLM-Mediated Pathology model (GLMP) (Figure 1). In contrast to standard PFMs that map a histology image patch directly to a numeric feature vector, our approach first generates a textual description of a given histology image patch using a pretrained general-purpose MLLM (e.g. Gemini 2.5 Pro, Comanici et al. (2025)). A structured prompt guides the MLLM to describe only biological characteristics, such as cellular morphology and tissue architecture, and filter out non-biological information, such as stain colors and image brightness. The resulting text is then encoded into a final embedding using a pretrained general-purpose text encoder. This work demonstrates that compared with existing PFMs, GLMP prioritizes genuine biological differences in tissues over artificial technical artifacts due to TSIs, yielding more robust and consistent feature representations for downstream analysis.

## 2 RELATED WORK

**Advanced methods for mitigating batch effects in WSIs.**   Several advanced strategies integrate artifact mitigation directly into the model's architecture or training process. One such strategy is based on Generative Adversarial Networks (GANs), which learn an unsupervised image-to-image translation to harmonize stain appearance while preserving morphology (Zanjani et al., 2018; Shaban et al., 2019). Another approach adapts the Domain-Adversarial Neural Network (DANN) framework (Ganin et al., 2016) for histopathology, leveraging an auxiliary classifier with a gradient reversal layer to produce features that are task-predictive yet invariant to site-specific artifacts (Otálora et al., 2019; Lafarge et al., 2019). This process requires joint training with domain labels from multiple sites to enforce invariance. Recent work learns semantically invariant features by aligning histology images with paired pathology texts (Huang et al., 2023; Ikezogwo et al., 2023; Lu et al., 2024b). However, reliance on large-scale annotated pathology data makes these methods non-scalable given the scarcity and cost of such data.

**Application of MLLMs for pathology.**   Recent work has explored MLLMs for both direct classification and feature representation in pathology. Ferber et al. (2024) show that in-context learning enables general-purpose MLLMs such as GPT4 to outperform task-specific PFMs on tile-level tasks, but this approach does not provide task-agnostic numeric embeddings. The MLLM4PUE framework (Zhou et al., 2025) introduces an information bottleneck by prompting an MLLM to generate a single-word summary of an image and uses the word's hidden state as the embedding, but this method constrains the semantic representation to a single token, which severely limits the richness of the embedded histopathological information, and requires access to the MLLM's hidden state, which is not available for frontier proprietary models. In addition, these existing MLLM-based methods do not have mechanisms to mitigate batch effects.

## 3 MODEL

We introduce GLMP, a pathology model designed to generate generalizable patch-level histology embeddings that are robust to batch effects (Figure 1). In contrast to standard vision-only PFMs, GLMP leverages an MLLM to derive text representations that serve as a middleman between the image patch and the numeric embedding. More specifically, after partitioning a WSI into patches of size $128 \times 128 \, \mu\text{m}^2$ (approximately $256 \times 256$ pixels under $20\times$ magnification) using a non-overlapping sliding window, we use an MLLM (Gemini 2.5 Pro, Comanici et al. (2025) in our implementation) to generate text descriptions for the image patches. Guided by a structured prompt (Appendix E.1), the MLLM is instructed to generate outputs that focus on histopathologic features that are relevant to the underlying biological characteristics, yielding semantic descriptions that minimize non-biological, TSI-specific artifacts (see Appendix F for examples). These descriptions are processed by a text encoding model (Gemini Embedding, Lee et al. (2025) in our implementation) into numeric embeddings. See Appendix A for the technical details of the full workflow.

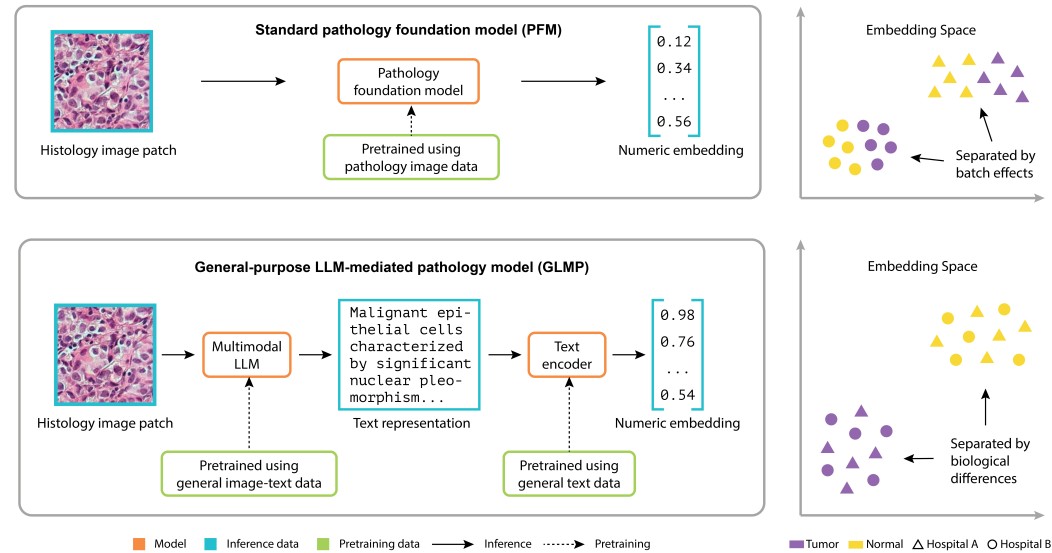

Figure 1: The GLMP framework. Instead of directly encoding an image patch, GLMP uses an MLLM (e.g. Gemini 2.5 Pro) to generate a semantic text description of the image's biological content, which is then converted to the final feature vector by a text encoding model (e.g. Gemini Embedding).

## 4 EXPERIMENTS

### 4.1 DATASETS AND BASELINES

We evaluated our approach on the following 5 datasets: CAMELYON16: Sentinel lymph node tissue samples (665586 patch-level images from 56 WSIs) with region-level annotations of metastatic tumor, acquired from 2 TSIs in the Netherlands (Radboud UMC, UMC Utrecht) (Bejnordi et al., 2017). TCGA-LUSC: Diagnostic slides (242740 patch-level images from 30 WSIs) of lung squamous cell carcinoma from The Cancer Genome Atlas (Network, 2012) with region-level annotations of invasive tumor (Loeffler & Kather, 2021), acquired from 3 TSIs: Mayo Clinic Rochester (MCR), the International Genomics Consortium (IGC), and Indivumed. AI4SkIN: Tissue samples (1426042 patch-level images from 60 WSIs) of cutaneous spindle cell neoplasms from 2 TSIs in Spain (HCUV and HUSC) (del Amor et al., 2025). TumSeg: WSIs (127690 patch-level images from 38 WSIs) of cutaneous squamous cell carcinoma from the Histo-Miner project (Sancéré et al., 2025), acquired from 2 TSIs in Cologne and Munich. MSBCD: Multi-Study Breast Cancer Dataset, a collection of five tissue samples (20110 patch-level images from 5 WSIs) curated from 3 independent breast cancer studies (Janesick et al., 2023; Andersson et al., 2021; 10x Genomics, 2022).

Our model is compared with a panel of state-of-the-art PFMs, including Virchow2 (Zimmermann et al., 2024), UNI2-h (Chen et al., 2024), hibou-L (Nechaev et al., 2024), H-optimus-1 (Dippel et al., 2024), Phikon-v2 (Hua et al., 2024), Prov-GigaPath (Xu et al., 2024a), and CONCH (image encoder only) (Lu et al., 2024b). We also include general-purpose vision models, such as DINOv2-base (Oquab et al., 2023) and ResNet-50 (He et al., 2016), as well as general-purpose MLLMs (image encoders only), such as Qwen2.5-VL-7B-Instruct (Wang et al., 2024) and Llama-3.2-11B-Vision (Grattonfiori et al., 2024). See Appendix C for baseline model details.

### 4.2 MULTI-TSI EMBEDDING CLUSTERING

To provide a visual overview of the robustness of GLMP compared with existing PFMs, we first demonstrate clustering results based on embeddings generated by GLMP and baseline PFMs on multi-TSI tissue slides in MSBCD. Clustering is performed at the patch level using $k$-means. As shown in Figure 2, GLMP clusters are consistently aligned with biological tissue types (e.g., brown for cancer, blue for stroma, red for adipose). The clusters are well-distributed across tissue slides

and TSIs, indicating robustness against TSI-specific signatures. In contrast, embeddings by UNI2-h, the most downloaded pathology foundation model on HuggingFace (334k as of September 2025), are predominantly driven by batch effects, as the three TSIs (with the exception of some noise in the image backgrounds) have three mutually exclusive sets of clusters (e.g., green and brown for TSI 1; blue, red, purple for TSI 2; orange for TSI 3). Similar performance is observed in other baseline PFMs (Appendix D.2). These results show that the GLMP representation prioritizes genuine biological features over technical artifacts in tissues.

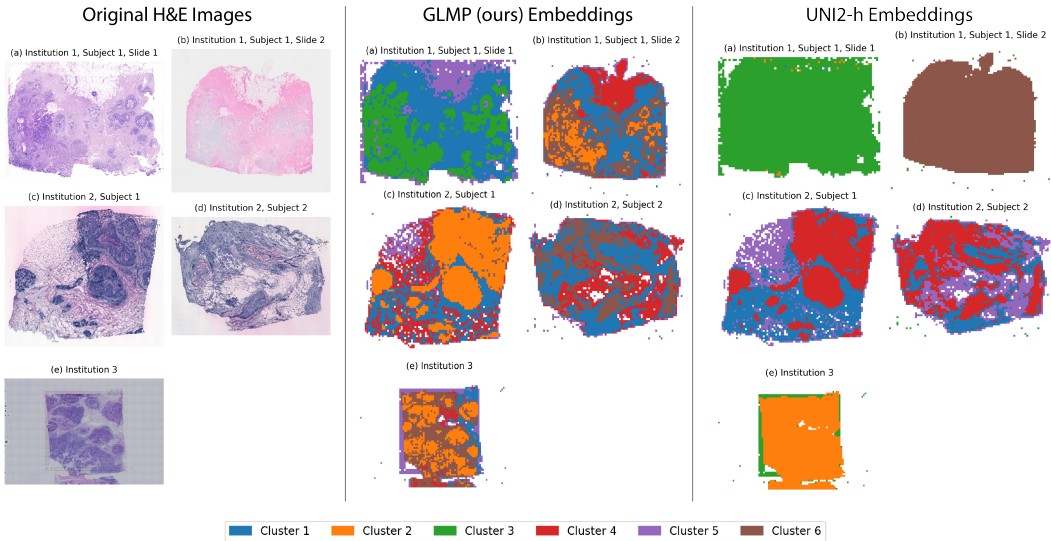

Figure 2: Clustering results using $k$-means on histology image patch embeddings generated by different models, along with the original H&E images.

### 4.3 VISUALIZATION OF THE EMBEDDING SPACE

To better understand the embedding space of different models, we perform principal component analysis (PCA, Bishop & Nasrabadi (2006)) and t-SNE (Maaten & Hinton, 2008) to visualize the patch-level embeddings from CAMELYON16 and TCGA-LUSC. The resulting 2D projections are colored by both tissue class and TSIs to assess whether the latent structure reflects biological signals or batch effects. The dimension reduction results show that GLMP learns representations that differentiate distinct tissue types while remaining robust with respect to TSI-specific signatures (Figure 3 & Appendix D.2). Across both datasets, GLMP embeddings show good separation between tumor and normal patches. When colored by TSI, the points are intermingled, suggesting minimal influence by TSI-specific signatures. By comparison, for the embeddings produced by baseline PFMs such as UNI2-h, hibou-L, and Phikon-v2, patches from different TSIs are clearly separated, and distances between TSIs tend to be larger than those between tissue classes. Together, these findings demonstrate that proximity in GLMP's embedding space primarily reflects biological similarity rather than TSIs, while baseline PFMs are much more affected by batch effects.

### 4.4 CROSS-TSI GENERALIZATION

An important clinical application of PFMs is the prediction of tissue classes. In practice, models are commonly trained on data from one or more TSIs and then deployed to new TSIs. Thus, we assess the cross-TSI generalization of GLMP compared with baseline PFMs by performing cross-TSI testing on the patch-level tissue classification task using CAMELYON16 and TCGA-LUSC. In each split of the cross-TSI testing, one TSI is selected to produce the testing image patches, while patches from the other TSIs are used for training. Embeddings are generated by the model for all the patches, followed by linear probe training to classify tumor vs. normal. We also performed within-TSI prediction for the same task, where patches from all the TSIs are mixed together and randomly sampled to form the training-testing split, so as to assess how the model's performance is

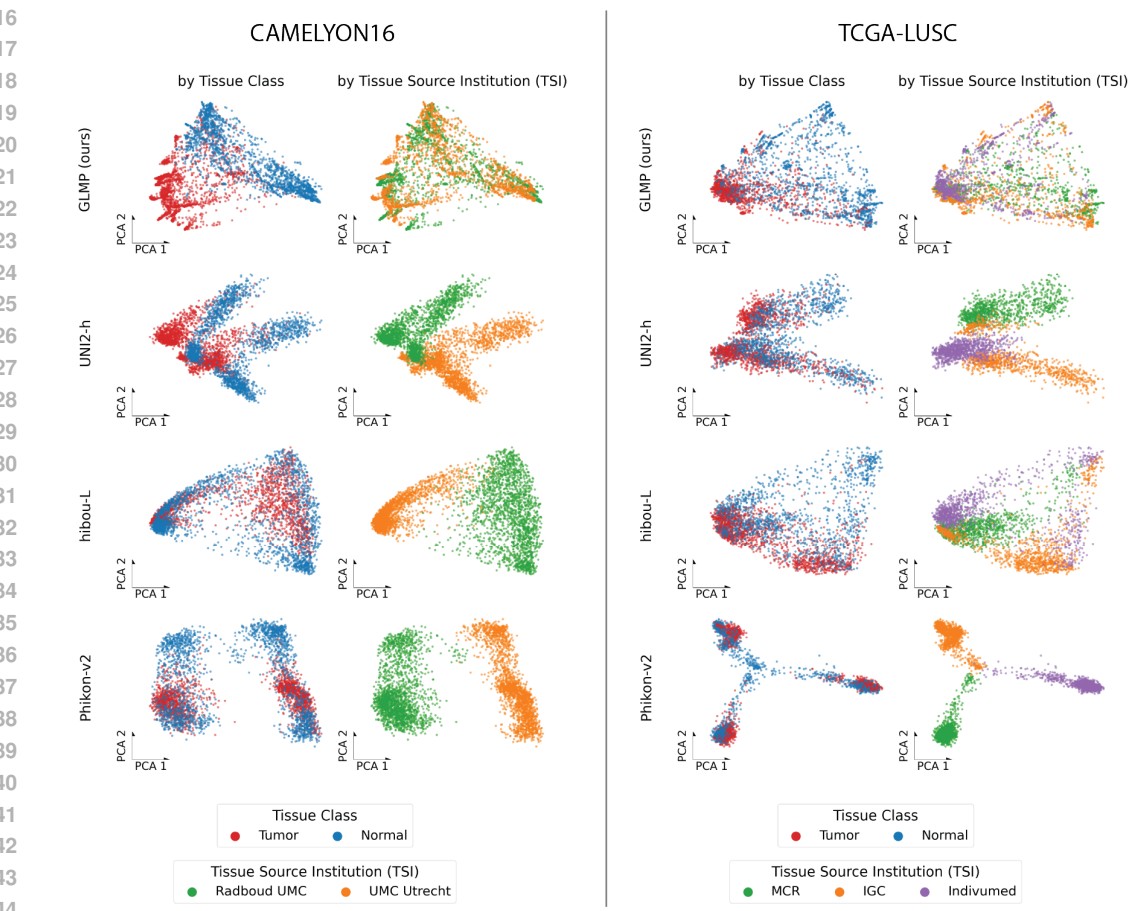

Figure 3: PCA visualization of embeddings generated by GLMP and baseline PFMs on CAME-LYON16 and TCGA-LUSC datasets, grouped by tissue class and TSI.

affected when testing is done on external instead of internal TSIs. Full experimental protocols are provided in Appendix B.

The generalization results show that GLMP produces highly discriminative embeddings (Figure 4). On CAMELYON16, most models achieve a near-perfect accuracy, and GLMP's performance is comparable with the baseline PFMs, which is notable given that GLMP has not been pretrained on large-scale histology-specific data. On TCGA-LUSC, the overall accuracy for all models is lower, which indicates that this dataset is more challenging than CAMELYON16. While some baseline PFMs still achieve a higher accuracy than GLMP for within-TSI prediction, GLMP outperforms all baseline models in cross-TSI generalization, which highlights its robustness to TSI-specific artifacts. While all baseline PFMs show a substantial performance drop when testing on external TSIs, GLMP maintains its performance, demonstrating that its embeddings are generalizable across domain shifts in TSIs. Compared with GLMP, general-purpose vision models (e.g. DINOv2-base, ResNet-50) and the vision encoders of general-purpose MLLMs (e.g. Qwen2.5-VL-7B-Instruct, Llama-3.2-11B-Vision) show significantly lower performance in both within-TSI and cross-TSI testing, which indicates that these broad-domain models have limited applicability to histology images without further domain-specific adaptation. In both datasets, compared with the baseline PFMs, GLMP has the least reduction in performance between within-TSI and cross-TSI evaluation, which in practice means that GLMP can be deployed to new TSIs without the need for costly model retraining or fine-tuning. The fact that GLMP's advantage is more pronounced on the more challenging TCGA-LUSC dataset suggests that GLMP's language-mediated representations are particularly effective at capturing robust biological features when the signal-to-noise ratio is lower.

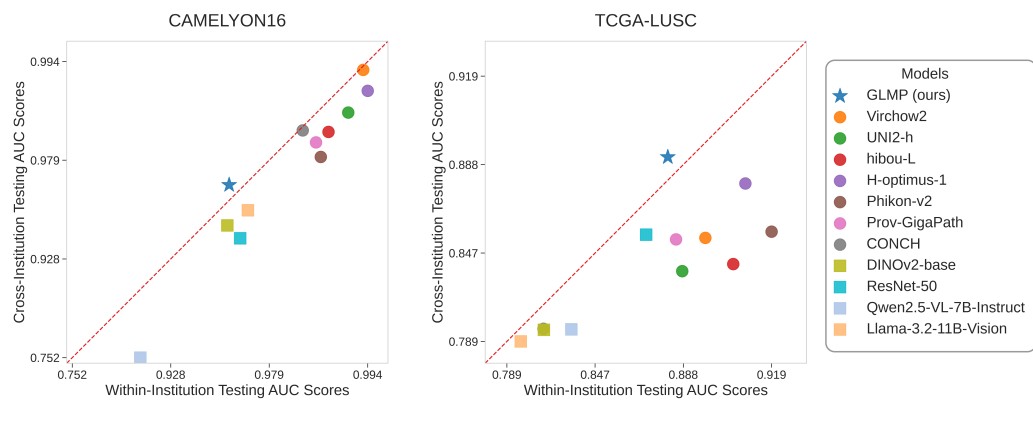

Figure 4: Accuracy of within-TSI testing (i.e. training data includes some WSIs from the testing TSIs) vs. cross-TSI testing (i.e. training does not include any WSIs from the testing TSIs) on CAMELYON16 and TCGA-LUSC tissue classification tasks. In both setups, patches in training and testing sets are from disjoint WSIs, ensuring that patches from the same WSI do not appear in both sets. Points falling below the $y = x$ line indicate a performance drop in model generalization when new TSIs (not included during training) are used for testing. Accuracy is measured by AUC, with the axes transformed using -log(1-AUC) for clearer visualization scale. PFMs are marked with circles while general-purpose vision models are marked with squares.

### 4.5 ROBUSTNESS TO TSI CONFOUNDING

A particularly dangerous case of batch effects is TSI confounding, where TSI-specific artifacts are correlated with true biological signals. Models might exploit these spurious correlations as predictive shortcuts, leading to biased predictions on new TSIs where the correlations do not hold. This kind of shortcut learning has been found to be a common pitfall for PFMs and natural vision models (Kömen et al., 2024; Hermann et al., 2023). To evaluate the robustness of GLMP against TSI confounding, we follow a protocol adapted from Kömen et al. (2024) to create training sets with increasing correlations between the tissue class and TSI using a CAMELYON16 subcohort. Model performance is evaluated on a test set constructed with the direction of correlation reversed (Table 1). This setup simulates a challenging real-world scenario where a model trained on biased data needs to generalize to an unbiased or reverse-biased setting. Since TSI-specific biases in WSIs are commonly addressed by stain normalization (Kömen et al., 2024; Lin et al., 2025; Nguyen & Ho, 2025; Yun et al., 2024), we provide baseline PFMs with two widely used stain normalization techniques: Reinhard (Reinhard et al., 2001) and Macenko (Macenko et al., 2009).

Table 1: Dataset splits for the TSI confounding experiment on CAMELYON16. The training sets introduce spurious correlations between the tissue class and TSI, while the testing set is constructed with a reversed correlation.

| Split | Set | Radboud UMC | | UMC Utrecht | | Total Patches |
|---|---|---|---|---|---|---|
| | | # Normal | # Tumor | # Normal | # Tumor | |
| 50/50 (no bias) | Training | 1,600 | 1,600 | 1,600 | 1,600 | 6,400 |
| 75/25 (low bias) | Training | 2,400 | 800 | 800 | 2,400 | 6,400 |
| 100/0 (high bias) | Training | 3,200 | 0 | 0 | 3,200 | 6,400 |
| | Testing | 0 | 800 | 800 | 0 | 1,600 |

Figure 5 shows that GLMP is robust to TSI-based batch effects as a confounder, maintaining stable performance across all training conditions. Even in the most challenging scenario where the training and testing data are fully biased in the opposite directions, GLMP shows only minimal reduction in prediction accuracy, indicating that it effectively captures biological signals over TSI-specific

artifacts when learning to predict tumor status. In contrast, without stain normalization, baseline PFMs perform well on the no-bias and low-bias splits but show severe degradation on the high-bias split. Their AUCs become lower than 0.3, with some near zero, suggesting that they rely on TSI signatures as a shortcut learning proxy for the tissue class. When Reinhard normalization is applied to the baseline PFMs, their performance (with one exception) remains worse than random guessing. With Macenko normalization, the baseline PFMs show improved performance but they remain below GLMP. These findings indicate that while stain normalization helps mitigate batch effects across TSI for baseline PFMs, it does not fully resolve the issue. On the other hand, GLMP's language-mediated representations are less susceptible to spurious correlations in the training data, enabling consistent generalization without the need for normalization. GLMP's generalizability under severely biased data conditions highlights its potential for reliable deployment in real-world clinical settings where data biases are common.

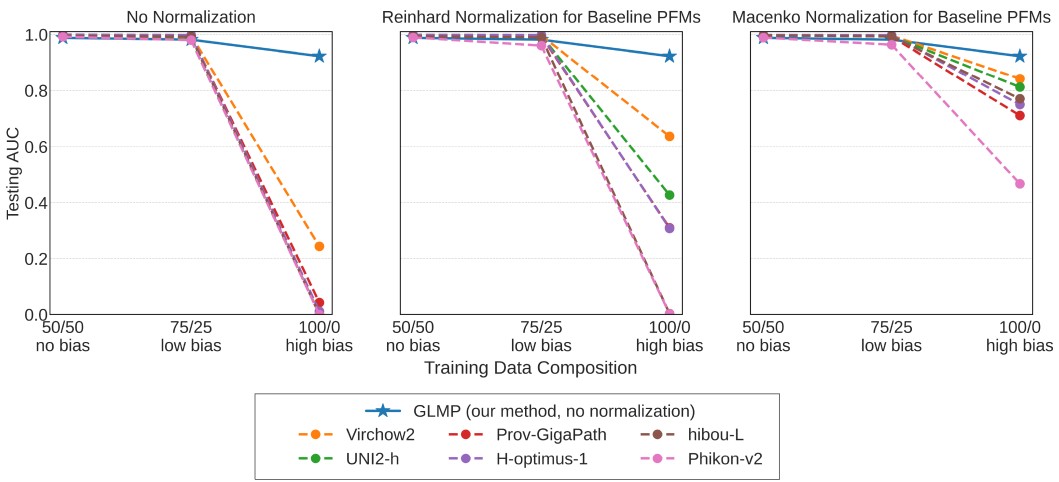

Figure 5: Tissue classification performance under increasing correlations between class label and TSI in CAMELYON16. A single testing set is used for all training conditions, which has the opposite direction of label-TSI correlation to the biases in the 75/25 and 100/0 training sets. Baseline PFMs are given the option of using stain normalization techniques (Reinhard and Macenko).

## 4.6 TSI PREDICTION

The ideal PFM should be invariant to non-biological technical variation and agnostic to TSIs. To quantify TSI-specific batch effects, we adopt the TSI prediction task as in Kömen et al. (2024). Specifically, for each model, we train a linear probe to predict a patch's TSI from its embedding and evaluate its performance via cross-TSI testing (Appendix B). Lower TSI prediction accuracy indicates less potential influence by TSI-specific artifacts and thus better robustness to batch effects.

Most baseline PFMs achieve very high TSI prediction accuracy, often close to 100%, indicating that their embeddings contain strong TSI-specific signatures. In contrast, GLMP's TSI prediction accuracy is only slightly above random chance, suggesting that GLMP's embeddings are largely free of information about TSIs (Table 2). The results confirm that GLMP is more robust to batch effects and less likely to exploit TSI-specific artifacts as predictive shortcuts than existing PFMs.

We further assessed the preeminence and concentration of TSI-specific signatures among all the variations captured in the embeddings. This is done by examining the predictability of TSI by the top principal components (PCs). A KNN classifier is trained on the top PCs of the embeddings based on a protocol adapted from Kömen et al. (2024). Since the top PCs reflect the directions of highest variance in the embeddings, this experiment evaluates how prominent the TSI-specific signatures are in the model feature space. As shown in Figure 6, baseline PFMs achieve high TSI prediction accuracy with only a few PCs, indicating that non-biological artifacts are among the dominant sources of variation in their embeddings. This pattern holds true even when baseline PFMs are assisted with stain normalization. In contrast, GLMP's TSI prediction accuracy remains

Table 2: Model susceptibility to batch effects, measured by accuracy on the TSI prediction task using a linear probe. Lower accuracy indicates less potential influence by TSI-specific signatures and higher robustness to technical artifacts, with a truly TSI-agnostic model expected to perform no better than random chance.

| Model | CAMELYON16 | TCGA-LUSC | AI4SkIN | TumSeg |
|---|---|---|---|---|
| Random chance (ideal performance) | 0.5000 | 0.3333 | 0.5000 | 0.5000 |
| GLMP (ours) | 0.6212 | 0.4750 | 0.5287 | 0.6154 |
| Virchow2 | 0.9998 | 0.9600 | 0.9926 | 0.9209 |
| UNI2-h | 0.9999 | 0.9690 | 0.9965 | 0.9201 |
| hibou-L | 0.9995 | 0.9593 | 0.9983 | 0.8962 |
| H-optimus-1 | 0.9999 | 0.9794 | 0.9914 | 0.8896 |
| Phikon-v2 | 0.9999 | 0.9973 | 0.9894 | 0.9027 |
| Prov-GigaPath | 0.9997 | 0.9560 | 0.9893 | 0.9278 |
| CONCH | 0.9985 | 0.7222 | 0.9536 | 0.8559 |
| DINOv2-base | 0.9966 | 0.7335 | 0.9640 | 0.8395 |
| ResNet-50 | 0.9877 | 0.8316 | 0.9690 | 0.8030 |
| Qwen2.5-VL-7B-Instruct | 0.9990 | 0.8124 | 0.9907 | 0.8825 |
| Llama-3.2-11B-Vision | 0.9976 | 0.6665 | 0.9813 | 0.8467 |

close to random chance even when using a high number of PCs. These results show that GLMP encodes substantially less TSI-specific signatures than existing PFMs.

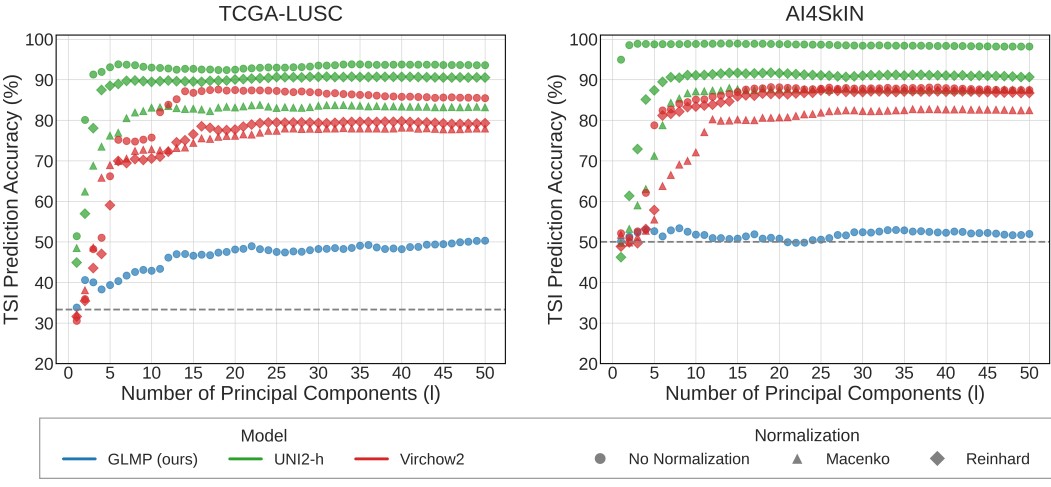

Figure 6: Accuracy for predicting TSI in TCGA-LUSC and AI4SkIN using KNN based on the top principal components (PCs) of the embeddings generated by different models. The horizontal dashed line indicates random chance.

## 4.7 ABLATION STUDIES

To better understand GLMP's robustness to batch effects and its downstream performance, we conduct ablation studies to investigate the contribution of different components in the model.

First, we would like to understand why GLMP is robust to TSI-specific signatures. We hypothesize that the biology-focused prompt for the MLLM filters out technical artifacts in the text representations of the images. To validate this hypothesis, we modify the prompt to explicitly instruct the MLLM to comment on non-biological characteristics, such as staining intensity and color profile, in addition to biological features (Appendix E.2). Then the resulting embeddings are applied to the TSI prediction task on CAMELYON16. As shown in Table 3, when the text representations are polluted with descriptions about non-biological attributes, the resulting embeddings become much more predictive of TSI. This result indicates that by focusing the MLLM on biological features, GLMP effectively filters out technical artifacts, making the embeddings more robust to batch effects

across TSIs. The language mediation layer in GLMP serves as a gatekeeper that determines what information is allowed to pass from the image to the embeddings based on the instructions in the MLLM prompt.

Table 3: Effects of MLLM prompt focus on TSI prediction on CAMELYON16. Lower scores indicate better robustness against batch effects.

| MLLM Prompt Focus | TSI prediction accuracy |
|---|---|
| Biological features only | 0.62 |
| Biological + non-biological features | 0.73 |

Next, given the rapid evolution of MLLMs, we investigate how the choice of MLLM affects GLMP's performance. We replace Gemini 2.5 Pro, the default MLLM in GLMP, with Gemini 2.0 Flash, a faster model with lower reasoning strength and overall intelligence. Both implementations are applied to the CAMELYON16 tissue classification task. As shown in Table 4, using Gemini 2.0 Flash leads to a drop in classification accuracy compared with Gemini 2.5 Pro. This finding indicates that the MLLM's overall capability does impact GLMP's performance.

Finally, since MLLMs play a vital role in GLMP, it is natural to ask whether the MLLM itself can be directly applied to pathology tasks without the language-mediated pipeline. We compare GLMP with the MLLM-only approach, i.e. direct tissue classification using Gemini 2.5 Pro on CAMELYON16, where the MLLM is prompted to respond whether the tissue in a given image patch is tumor or normal (Appendix E.3). As shown in Table 4, GLMP outperforms the MLLM-only approach, which indicates that MLLMs alone are insufficient for achieving optimal performance on pathology tasks.

Table 4: Tissue classification accuracy of different MLLM-based methods.

| Method | Tissue classification accuracy |
|---|---|
| GLMP (Gemini 2.5 Pro) | 0.91 |
| GLMP (Gemini 2.0 Flash) | 0.74 |
| MLLM-only (Gemini 2.5 Pro) | 0.70 |

## 5 CONCLUSION

In this work, we have introduced GLMP, a pathology model with robustness to batch effects and generalizability across TSIs. A core innovation of GLMP is the use of an MLLM to translate histology images into biology-focused text intermediaries before encoding them numerically. A key finding in our experiments is that when used properly, vision-language models pretrained on massive general-purpose data can effectively extract pathologically relevant features. Our model provides a new paradigm for pathology modeling that leverages the rapid advancements in general-purpose MLLMs instead of relying solely collecting large-scale histology-specific data.

A limitation of this work is that GLMP is evaluated only on H&E-stained WSIs. Future work will explore the applicability of GLMP to other staining and imaging modalities, such as immunohistochemistry immunofluorescence, and special histochemical stains. In addition, the current implementation is not specialized for specific diseases. Future work will explore tailoring GLMP for disease-specific applications in clinical settings. Furthermore, the current study is limited to empirical validation of GLMP's robustness to batch effects. Theoretical analysis of GLMP is needed to formalize the conditions under which GLMP can effectively generalize across TSIs.

## ETHICS STATEMENT

This study relies exclusively on publicly released datasets and pretrained models. No human participants are involved. The authors declare no conflicts of interest for this work and are not aware of

any violation of relevant ethical standards, privacy policies, security protocols, legal requirements, or research integrity guidelines.

## REPRODUCIBILITY STATEMENT

The code for this project is provided in the Supplementary Materials and will be released on GitHub upon paper acceptance. Experimental details, including data processing steps and model parameter settings, are provided in the Appendix to facilitate reproducibility.

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

LLM USAGE DISCLOSURE

Large language models are used for coding assistance and manuscript editing. The authors have reviewed all the outputs and take full responsibility for the content.

## A GLMP IMPLEMENTATION DETAILS

**Foreground tissue detection.** Each whole-slide image (WSI) is tiled into non-overlapping square patches of a fixed physical size (e.g., $128 \times 128 \, \mu m$). We derive an adaptive tissue mask by converting a downsampled WSI thumbnail to the Hue, Saturation, and Value (HSV) color space and applying Otsu's method to the non-zero values of each channel (Schreiber et al., 2024). Patches are retained for subsequent analysis only if their masked tissue area exceeds a minimum threshold of 1%.

**Patch grouping for efficient MLLM prompting.** We use a group-based strategy to ensure efficiency of the MLLM API calls. For each WSI, foreground patches are first clustered into $k$ histology patch groups using $k$-means based on their similarity in the feature space of a vision encoder backbone. Since within-slide clustering analysis by definition is invariant to to slide-level (and thus TSI-level) batch effects, which are constant for any single tissue slide, this clustering step is commonly used for efficiently organizing within-slide patches into histopathologically representative groups Ding et al. (2024); Wang et al. (2023); Shaban et al. (2025). Existing works find this single-slide clustering process to be stable with respect to model choice and the number of clusters when $k$ is between 10 and 25 per WSI (Claudio Quiros et al., 2024; Alfasly et al., 2025; Zhang et al., 2024; Shaban et al., 2025; Ge et al., 2025). We thus set the number of clusters to $k = 10$ in our experiments to control MLLM context window usage, since long contexts can lead to performance degradation (Liu et al., 2023; An et al., 2024; Jha et al., 2024). The clustering serves only as a preprocessing step for efficient API usage and does not contribute to the final embeddings, which are generated entirely through the MLLM-text encoder pipeline. After clustering, a soft probabilistic membership score is then computed for each patch to every cluster via a temperature-scaled softmax over their cosine similarities:

$$p_{ij} = \frac{\exp(\langle \hat{\mathbf{x}}_i, \hat{\boldsymbol{\mu}}_j \rangle / \tau)}{\sum_{\ell=1}^{k} \exp(\langle \hat{\mathbf{x}}_i, \hat{\boldsymbol{\mu}}_\ell \rangle / \tau)} \tag{1}$$

where $\langle \cdot, \cdot \rangle$ denotes cosine similarity and $\tau$ is the softmax temperature, which is set to $\tau = 0.1$ as recommended in temperature-tuning literature (Agarwala et al., 2020; Ferrante et al., 2023). These probabilities are used to select $m = 25$ representative patches per cluster for the MLLM API call.

**Semantic description generation.** Biologically focused semantic descriptions are generated for all histology patch groups in an API call to the MLLM (default: Gemini 2.5 Pro (Comanici et al., 2025)). The request consists of a structured prompt (Appendix E.1) followed by representative patches from each group, with images placed immediately after their textual identifiers. To induce a deterministic and structured output, the generation temperature is set to 0.0, and the model is instructed to return a single JSON object covering all groups. The process is repeated 10 times to form an ensemble of outputs, allowing minor textual variations from the inherent stochasticity of the MLLM to be averaged out in the subsequent embedding step. This stage yields 10 JSON objects, each containing a complete description of all histology patch groups.

**Patch-level embedding generation.** The structured MLLM descriptions for the $k$ histology patch groups are encoded into $k$ corresponding semantic embeddings using a text embedding model (Gemini Embedding (Lee et al., 2025)). To reduce randomness in outputs due to the inherent stochastic properties of MLLMs, the embeddings from 10 repeated MLLM runs for each group are then averaged to yield a single embedding $\mathbf{s}_j$ for each group $j$. The final GLMP embedding for an individual patch, $\mathbf{z}_i$, is then computed as a weighted sum of these $k$ group embeddings using the patch's soft probabilistic membership score $p_{ij}$ from (Equation 1) as the weight:

$$\mathbf{z}_i = \sum_{j=1}^{k} p_{ij} \mathbf{s}_j \tag{2}$$

## B   EXPERIMENTAL DETAILS

**Data preprocessing.**   For all datasets, image patches are extracted to represent a physical area of $128 \times 128\,\mu$m at an effective 20x magnification ($\approx 0.5\,\mu m/pixel$). The exception is the Tum-Seg dataset, which is available only at a low resolution (1.25x), from which we extract 256µm x 256µm patches. For CAMELYON16 and TCGA-LUSC, we select foreground patches located entirely within an annotated region ("Tumor") or entirely outside of any annotated region ("Normal").

**Linear probe and training.**   We train a single linear layer on frozen embeddings, using the Adam optimizer with a learning rate of $10^{-4}$ and cross-entropy loss. Training is performed with a batch size of 256 patches for 20 epochs, and we use the weights from the final epoch for evaluation.

**Evaluation protocols.**

- **Cross-TSI testing:** We evaluate generalization across TSIs by training and testing on slides from different TSIs. For this task, patch-level datasets are constructed by randomly sampling up to 2,000 tumor or normal patches from each slide. On the CAMELYON16 dataset, which includes 2 TSIs, models are trained on one TSI and tested on the other, and vice-versa. For the TCGA-LUSC dataset, we employ a leave-one-TSI-out testing, training on two TSIs and testing on the third, with the held-out TSI in rotation.

- **Within-TSI testing:** We pool slides from all TSIs and use 5-fold splitting for creating the training and testing sets. Splits are created at the slide level so that no slide appears in more than one fold, and stratification maintains a balanced distribution of TSIs across folds. The patch sampling procedure (up to 2,000 patches per slide) is identical to that used in the cross-TSI experiments.

- **TSI Confounding:** We adopt the protocol from Kömen et al. (2024) using a 56-slide subcohort from CAMELYON16. From each slide, 200 patches are sampled, with tumor patches drawn from metastatic slides and normal patches only from non-metastatic slides. This design yields three training splits with increasing levels of label–site correlation (50/50 no bias, 75/25 low bias, and 100/0 high bias), as summarized in Table 1. A linear probe is then trained on each biased split and evaluated on a test set constructed with the opposite correlation.

- **TSI Prediction:** We quantify site-specific artifacts by training a linear probe to predict the acquisition site across four datasets: CAMELYON16, TCGA-LUSC, AI4SkIN, and TumSeg. For each dataset, we sample up to 2,000 patches per slide and perform a 5-fold combined-TSI cross-testing. We report the mean prediction accuracy and standard deviation across folds.

- **k-NN on Principal Components:** Following the protocol of Kömen et al. (2024), we evaluate site-specific signal in the TCGA-LUSC and AI4SkIN embeddings. Using 5-fold cross-testing at the slide level, we train a $k$-nearest neighbors classifier ($k = 5$, cosine similarity) on the top $l$ principal components, with $l$ varying from 1 to 50. This assesses how much site information is captured in the dimensions of highest variance.

# C BASELINE MODEL CONFIGURATIONS

Our comparative analysis involved 12 foundation models, encompassing pathology-specific vision encoders, general-purpose vision models, and MLLMs (Table 5). All pre-trained models were obtained from the Hugging Face Hub, except for the ResNet-50 baseline from torchvision. For each model, the feature extraction procedure was tailored to its specific architecture and the authors' recommendations. For Virchow2, we concatenated the classification token with the mean-pooled representation of all patch tokens. For hibou-L, we utilized the model's `pooler_output`, while for Phikon-v2 and the DINOv2 vision encoder, we extracted the final hidden state of the classification token. For UNI2-h, H-optimus-1, and the Prov-GigaPath tile encoder, we used the single feature vector returned directly by a forward pass. For the contrastive vision-language model CONCH, we obtained features from its vision encoder prior to the final contrastive projection layer. The ResNet-50 baseline was represented by its final global average pooling layer. For the MLLMs, we investigated two types of semantic embeddings: features from the vision projection layer, and contextualized features from the full model's last hidden layer. Accordingly, for Qwen2.5-VL-7B-Instruct, we obtained the embeddings by mean-pooling the output of its `get_image_features` function. For Llama-3.2-11B-Vision, we obtained the contextualized embeddings by invoking the model's standard forward pass with a minimal, non-informative text prompt and then mean-pooling the representations from the last hidden layer.

Table 5: Overview of the foundation models benchmarked in this study.

| Model Repository ID | Architecture (Param.) | Pre-training Data | Dim. | Reference |
|---|---|---|---|---|
| **Pathology-Specific Vision Encoders** | | | | |
| paige-ai/Virchow2 | ViT-H/14 (632M) | 3.1M WSIs | 2560 | Zimmermann et al. (2024) |
| MahmoodLab/UNI2-h | ViT-H/14 (681M) | 350k WSIs | 1536 | Chen et al. (2024) |
| histai/hibou-L | ViT-L/14 (304M) | 1.14M WSIs | 1024 | Nechaev et al. (2024) |
| bioptimus/H-optimus-1 | ViT-g/14 (1.1B) | 1M WSIs | 1536 | Bioptimus (2025) |
| owkin/phikon-v2 | ViT-L/16 (304M) | 58.4K WSIs | 1024 | Filiot et al. (2024) |
| prov-gigapath/prov-gigapath | ViT-g/14 (1.1B) | 171k WSIs | 1536 | Xu et al. (2024a) |
| MahmoodLab/CONCH | ViT-B/16 (86M) | 1.17M image-text pairs | 512 | Lu et al. (2024b) |
| **General-Purpose Vision Encoders** | | | | |
| facebook/dinov2-base | ViT-B/14 (86M) | LVD-142M | 768 | Oquab et al. (2023) |
| torchvision/resnet50 | ResNet-50 (25.6M) | ImageNet-1K | 2048 | He et al. (2016) |
| **Multi-Modal Language Models (MLLMs)** | | | | |
| Qwen/Qwen2.5-VL-7B-Instruct | MLLM (7B) | Image-text/documents/video | 3584 | Wang et al. (2024) |
| meta-llama/Llama-3.2-11B-Vision | MLLM (11B) | 6B image-text pairs | 4096 | Grattonfiori et al. (2024) |

# D ADDITIONAL EXPERIMENTAL RESULTS

## D.1 MULTI-TSI EMBEDDING CLUSTERING

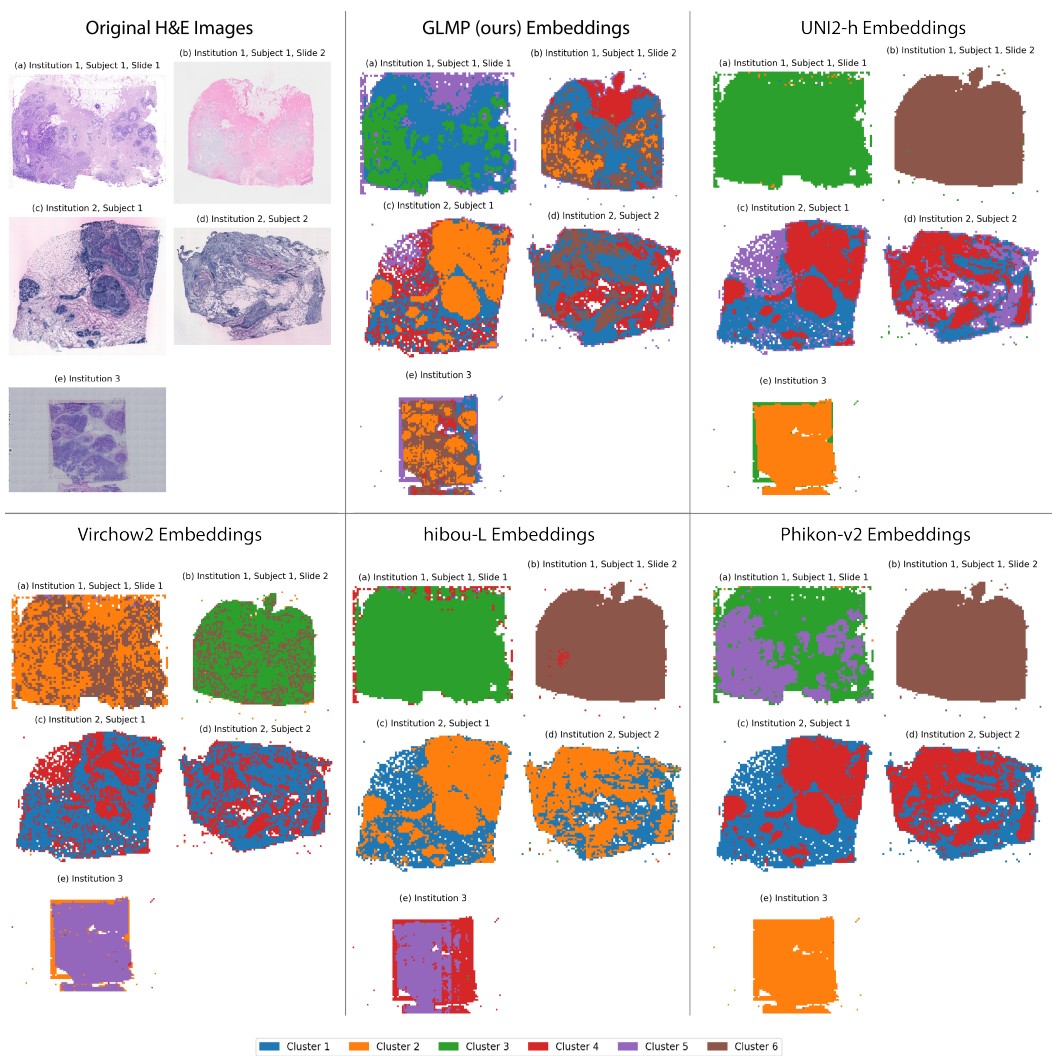

Figure 7: Clustering results on the MSBCD dataset using $k$-means on histology image patch embeddings generated by different models.

## D.2 VISUALIZATION OF EMBEDDING SPACES

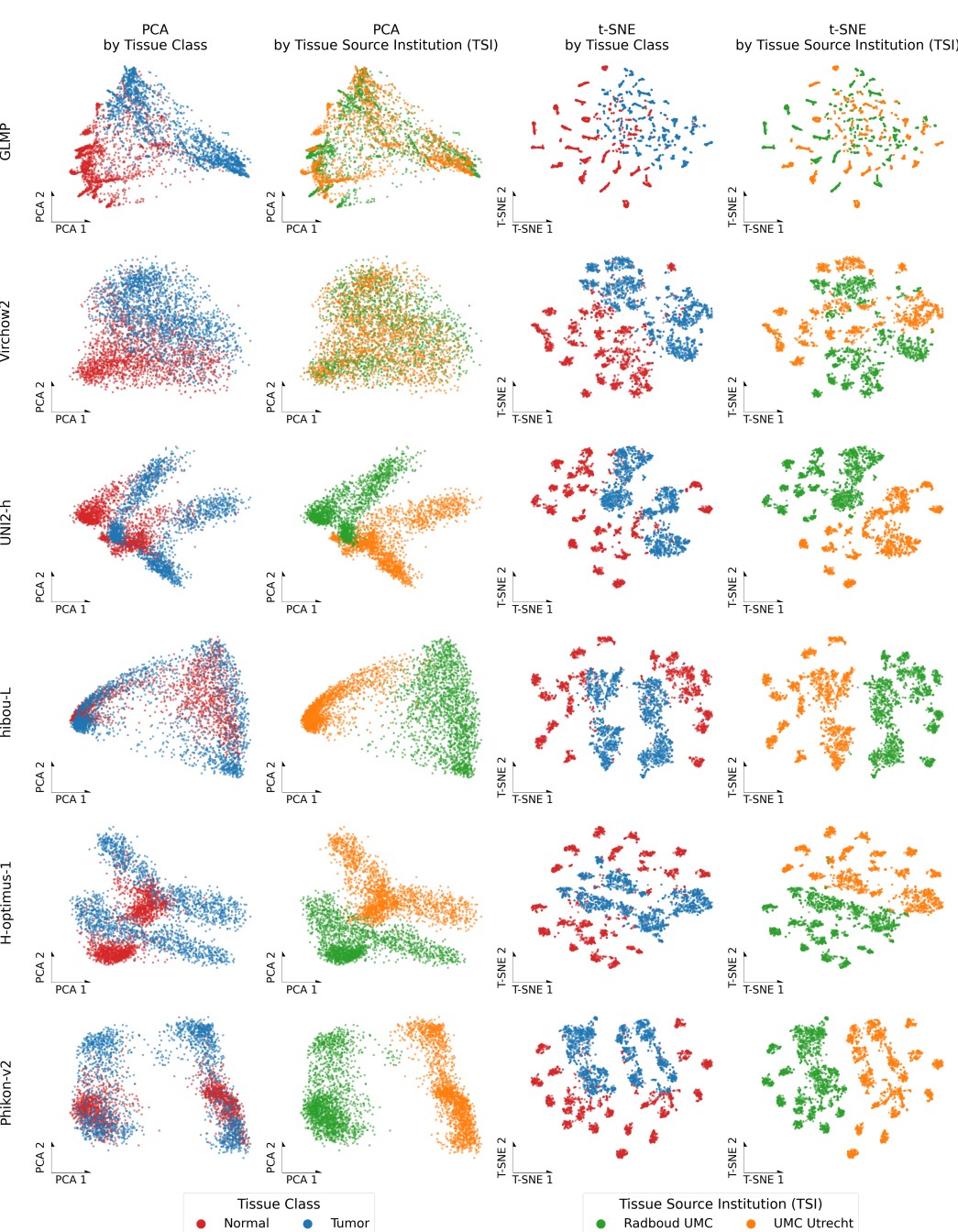

Figure 8: Dimension reduction of pathology model embeddings on the CAMELYON16 Dataset.

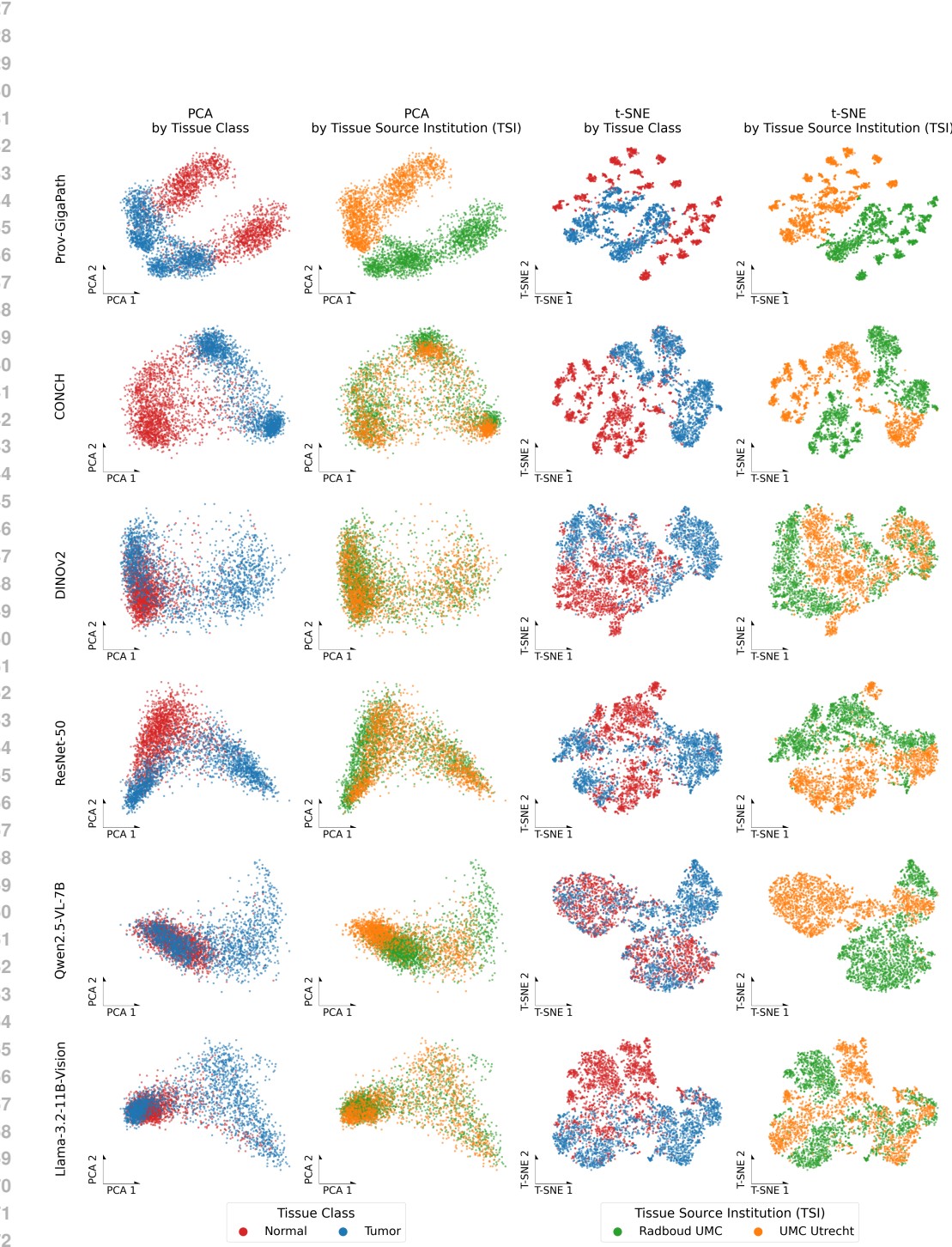

Figure 8: (Continued) Dimension reduction of pathology model embeddings on the CAMELYON16 Dataset.

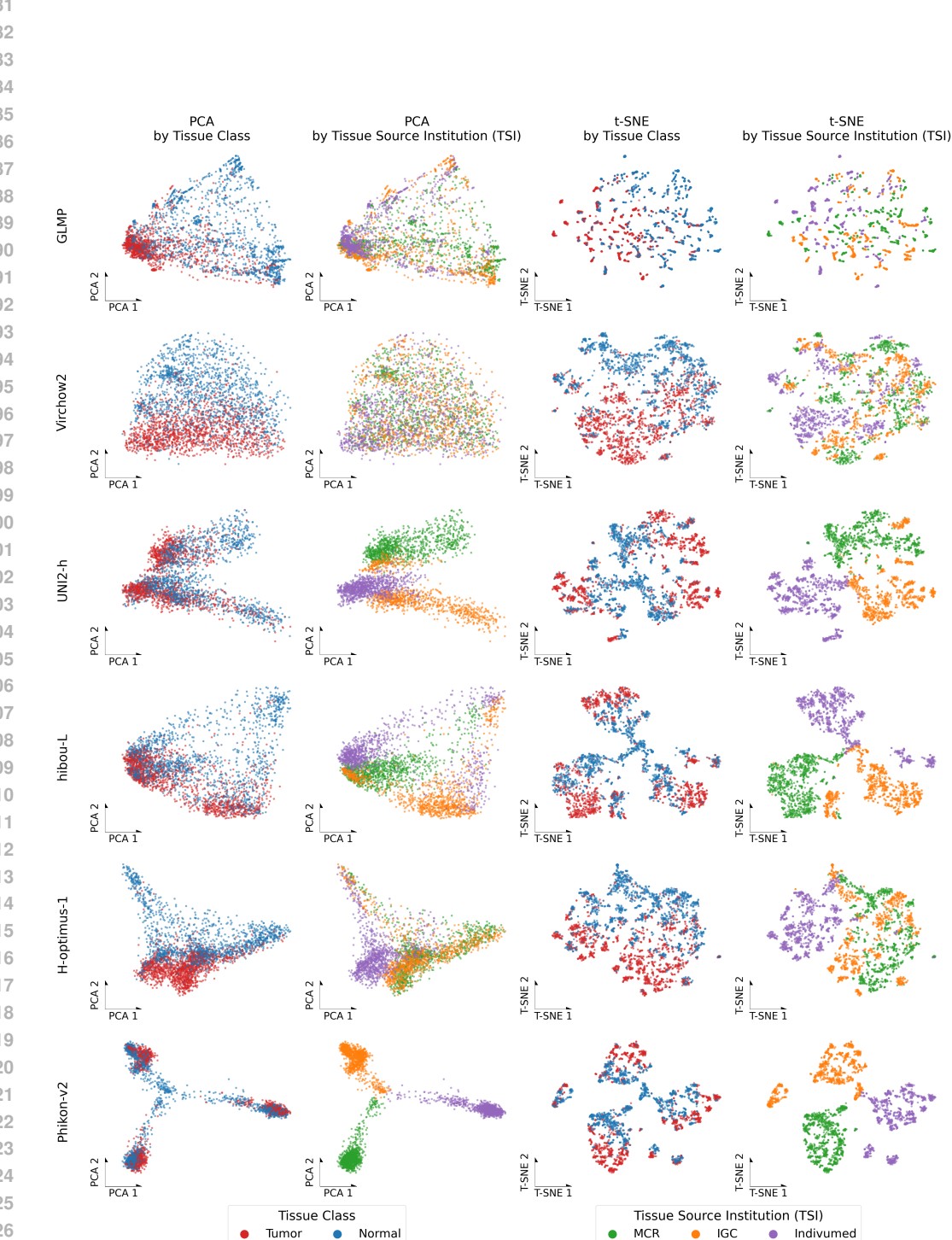

Figure 9: Dimension reduction of pathology model embeddings on the TCGA-LUSC Dataset.

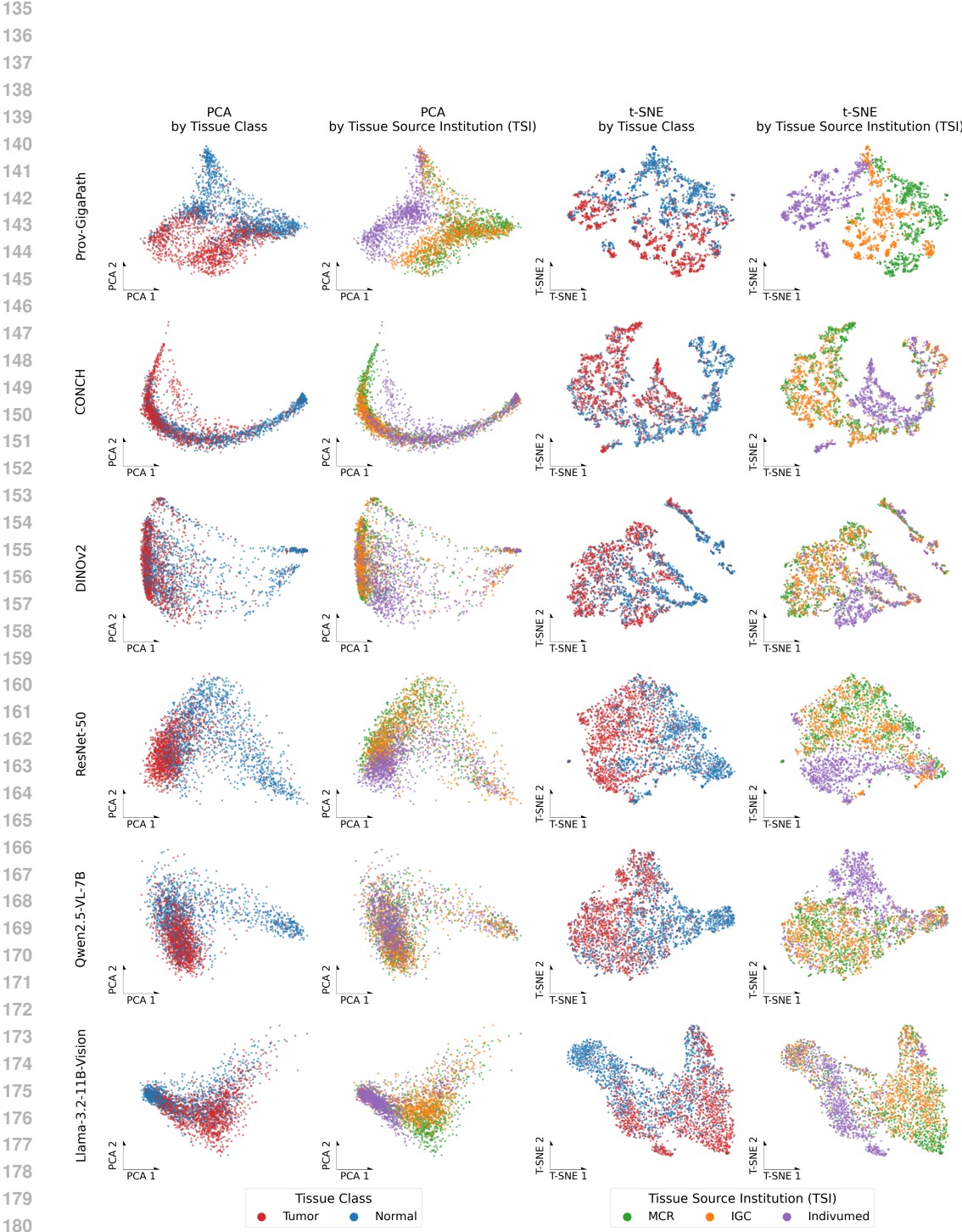

Figure 9: (Continued) Dimension reduction of pathology model embeddings on the TCGA-LUSC Dataset.

# E  MLLM PROMPT FOR GLMP

## E.1  STANDARD PROMPT

The following prompt guides the MLLM to generate biology-focused, artifact-free semantic descriptions for histology images, with __ORGAN__ and __SPECIES__ adapted to the specific dataset.

---

You are a board-certified __ORGAN__ pathologist.

**Input Structure & Order**
You will receive this entire set of instructions first. Following these instructions, a sequence of content will be provided:

1. A text line identifying a group, for example: "The subsequent image patches pertain to Group 1."

2. Immediately following this text, all image patches belonging to Group 1 will be provided.

3. This pattern will repeat for subsequent groups (e.g., "The subsequent image patches pertain to Group 2.", followed by its images, and so on for all available groups).

Your task is to process all groups and then generate a single JSON output summarizing each one.

**Context for Image Analysis**
The H&E image patches you will receive are from a single whole-slide image (WSI) of a __ORGAN__ tissue from __SPECIES__.

- **Group Delineation**: You must strictly use the textual group identifiers provided in the input stream (e.g., "The subsequent image patches pertain to Group 1.") to define which images belong to which group, the order of the groups, and for keying your final JSON output.

- **Intra-Group Similarity**: All patches within a specific group (as defined by its preceding textual identifier) are expected to share similar histologic features, though minor variations may exist.

- **Expected Content**: Depending on the specific tissue type indicated above, you may encounter a wide range of histologic components. The specific features will be pertinent to the organ/tissue system being examined.

**Tasks**

1. **Within-Group Synthesis**: For each group of images presented (e.g., those following "The subsequent image patches pertain to Group 1."), carefully examine **all** provided image patches belonging to that specific group. Synthesize a representative description that captures the **predominant, consistent, and defining** histologic features observed across these patches. For each group, your synthesis should specifically address the following aspects, which will directly correspond to the fields in the structured summary:

   - **Architectural Pattern**: Describe the predominant tissue arrangement (e.g., infiltrative growth, preserved native architecture, glandular formation, solid sheets) and structure density. If acellular or non-architectural, describe that arrangement.

   - **Cellular Morphology & Cytologic Grade**: Describe predominant cell features: relative cell size, cell density, nuclear pleomorphism, chromatin, nucleoli, mitotic activity, and cytoplasm. Assign an overall nuclear grade (low, intermediate, high) if applicable. If acellular, state 'Not applicable'.

   - **Key Structural Interface**: Describe the most significant structural boundary or architectural interface observed. Focus on the relationship between the main lesion and surrounding tissue, such as the status of a capsule, basement membrane, or the nature of a tumor-stroma interface. If not applicable, state so.

   - **Stromal Response & Inflammation**: Describe the stroma and the type, density, and location of any inflammatory infiltrate. If the group is predominantly stroma or inflammation, describe it here.

   - **Necrosis & Other Key Features**: Describe the presence and type of any necrosis. Report other key features only if they are diagnostically significant AND truly widespread across the majority of patches, thereby defining the group's overall character. Omit minor, focal, or incidental findings.

2. **Internal Comparison (Mental Step Only - Do Not Output)**: Mentally compare the synthesized features of each group against the others. This step is for you to refine your within-group synthesis (Task 1). Observing significant differences in key features between groups confirms they are distinct entities. Use this mental differentiation to ensure that the description for each group accurately and uniquely captures *its own* predominant characteristics. **Absolutely do NOT mention these mental comparisons or reference any other group in your final written output.** Each group's description must be entirely self-contained.

---

3. **Structured Summary (Final Output)**: After all groups and their images have been presented and analyzed, produce a **single JSON object**. This object will contain one top-level key for each group processed. The keys in the JSON (e.g., "Group 1", "Group 2") must exactly match the group numbering specified in the textual identifiers that introduced each image set (e.g., "Group 1" from "The subsequent image patches pertain to Group 1."). Base the descriptions for each field **strictly** on your synthesis from Task 1 for that specific group.

Example JSON structure:

```
{
  "Group 1": {
    "Architectural Pattern": "<description>",
    "Cellular Morphology & Cytologic Grade": "<description>",
    "Key Structural Interface": "<description>",
    "Stromal Response & Inflammation": "<description>",
    "Necrosis & Other Key Features": "<description>"
  },
  "Group 2": {
    // ... same structure ...
  }
}
```

**Style & Constraints for Output**

- **Focus on Predominant, Defining, and Widespread Features**: Your descriptions MUST reflect features that are predominant, consistent, and diagnostically significant for the entire group of images. Base descriptions on observations from the clear majority of patches. AVOID over-reporting minor or focal variations.

- **Absolutely NO Cross-Group References**: Under **NO circumstances** should the description for one group (e.g., Group 8) mention, compare itself to, or reference any other group. Each group's description MUST be entirely self-contained.

- **Do Not Mention Metadata in Descriptions**: Avoid mentioning patch size, number of patches, magnification, or the concept of clustering itself within the descriptive string values of the JSON.

- **Conciseness and Terminology**: Keep each field's description to 1-2 sentences. Be precise and use standard histopathology terminology.

- **Strictly Avoid Artifact Description**: Focus exclusively on biological features. Omit descriptions of technical artifacts unless they pervasively obscure the biological assessment of a group.

- **Strict Output Format**: Output **only** the JSON object—no extra introductory text, commentary, explanations, or apologies. The output must start directly with '{' and end directly with '}'.

The subsequent image patches pertain to Group 1.

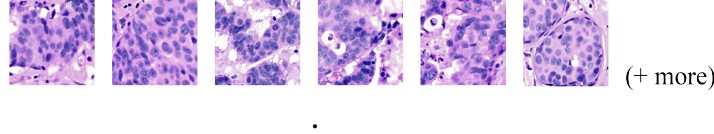

(+ more)

The subsequent image patches pertain to Group 2.

(+ more)

⋮

The subsequent image patches pertain to Group 10.

(+ more)

## E.2 PROMPT USED IN THE ABLATION STUDY

This prompt is identical to the biological-only prompt in Appendix E.1, except for the following additions and edits to request descriptions of non-biological artifacts in the ablation study.

---

**Addition to Task 1 fields**

- **Non-Biological Characteristics**: Describe consistent non-biological features that characterize the group, including stain color balance or strength, scanner focus or stitching artifacts, and site or batch cues. Report only if these characteristics are widespread within the group.

---

**Edits to Style and Constraints:** Replace "Strictly Avoid Artifact Description" with: "Report non-biological characteristics only when they are consistent and pervasive within the group. Do not over-interpret minor or focal artifacts."

---

**JSON Output**

```
{
  "Group 1": {
    "Architectural Pattern": "<description>",
    "Cellular Morphology & Cytologic Grade": "<description>",
    "Key Structural Interface": "<description>",
    "Stromal Response & Inflammation": "<description>",
    "Necrosis & Other Key Features": "<description>",
    "Non-Biological Characteristics": "<description>"
  },
  "Group 2": {
    // ... same structure ...
  }
}
```

---

## E.3 ABLATION: THE MLLM-ONLY APPROACH FOR TISSUE CLASSIFICATION

---

**Prompt for the MLLM-only approach for CAMELYON16 tissue classification**

You are a board-certified lymph node pathologist. Your task is to analyze a single H&E stained image patch provided to you. This patch is from a larger whole-slide image (WSI) of a lymph node.

Your analysis must be strictly confined to the features visible within this single patch. Do not make a diagnosis for the entire slide or the patient.

Based on your expert assessment of the morphology, classify the patch. Your response must be a single, valid JSON object and nothing else. The JSON object must contain a single key, "classification", and its value must be one of two exact strings: "Tumor" or "Normal".

For example, if you classify the patch as Normal, your **entire response** must be:

```
{
"classification": "Normal"
}
```

---

## F    EXAMPLES OF MLLM OUTPUT

Below are some examples of the structured descriptions generated by the MLLM (Gemini 2.5 Pro) for two histology patch groups from a WSI in MSBCD, using the biologically focused prompt in Appendix E.1.

Input Image (Group 1):

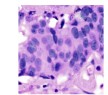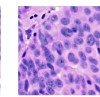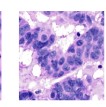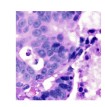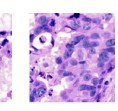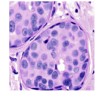 (+ more)

Generated Description (Group 1):

```
"Group 1":  {
  "Architectural Pattern":  "Disorganized, infiltrative pattern of mixed inflammatory
  cells within a fibrous and edematous stroma.  No discrete epithelial or glandular
  structures are identified.",
  "Cellular Morphology & Cytologic Grade":  "The cellular population is composed of
  small, mature lymphocytes with dark nuclei and scant cytoplasm, intermixed with
  larger histiocytes showing vesicular nuclei and more abundant cytoplasm.  Bland
  spindle cells are also present.  Not applicable for nuclear grade.",
  "Key Structural Interface":  "The inflammatory infiltrate diffusely permeates the
  stromal connective tissue without forming a distinct interface with a neoplastic or
  epithelial component.",
  "Stromal Response & Inflammation":  "The stroma is fibrotic and edematous, hosting
  a prominent, diffuse, mixed chronic inflammatory infiltrate of lymphocytes and
  histiocytes.",
  "Necrosis & Other Key Features":  "No overt necrosis is seen.  Scattered apoptotic
  bodies and cellular debris are present within the inflammatory milieu."
  }
```

Input Image (Group 2):

 (+ more)

Generated Description (Group 2):

```
"Group 2":  {
  "Architectural Pattern":  "Cohesive nests of neoplastic epithelial cells arranged
  in solid and cribriform patterns, consistent with ductal carcinoma in situ.  The
  nests fill and expand duct-like spaces.",
  "Cellular Morphology & Cytologic Grade":  "The neoplastic cells exhibit moderate to
  marked pleomorphism, with enlarged, hyperchromatic to vesicular nuclei, irregular
  nuclear contours, and prominent nucleoli.  The overall features correspond to an
  intermediate to high nuclear grade (Grade 2-3).",
  "Key Structural Interface":  "The neoplastic epithelial nests are sharply
  demarcated from the surrounding stroma, suggesting confinement within a
  ductal-lobular unit.",
  "Stromal Response & Inflammation":  "The periductal stroma is fibrous and contains
  a mild, patchy lymphoplasmacytic infiltrate.",
  "Necrosis & Other Key Features":  "Focal luminal necrosis is present, characterized
  by eosinophilic, granular debris.  Apoptotic cells are also noted within the
  epithelial nests."
  }
```

