# OpenReview forum: "LLM-mediated pathology models for robust cross-institution generalization"
_ICLR.cc/2026/Conference — ICLR 2026 Conference Withdrawn Submission_

### Official Review · Reviewer_Ggnd · 2025-10-22

**Soundness:** 2
**Presentation:** 4
**Contribution:** 3
**Rating:** 4
**Confidence:** 5

**Summary:**

This work presents GLMP, an LLM-based model for patch feature extraction, which generates patch-level features by using text embeddings from LLM-generated image captions. These LLM-based patch embeddings can then be used for downstream classification, demonstrating substantially-improved generalizability across tissue sites. While this model does not always achieve state-of-the-art, its consistency across tissue sites is impressive and a meaningful step towards improving clinical reliability. However, the paper currently evaluates on tasks with clearly defined morphological criteria, making its overall utility unclear. A wider range of tasks, such as molecular subtyping and survival prediction, would provide critical insight into which tasks this GLMP approach is appropriate.

Overall, I found the work interesting, as it introduces a relevant and unexplored approach for generating patch-level features with less tissue site bias. I would be willing to increase my score pending my comments described below.

**Strengths:**

There is substantial benefit to a patch foundation model with increased tissue site robustness. In current practice, this is primarily achieved by increasing institutional diversity within the model’s training set. Meanwhile, GLMP provides a VLM-agnostic method for obtaining highly generalizable patch embeddings without the need for self-supervised training.
The work compares against an extensive number of SOTA patch foundation models
The work examines performance across a satisfactory number of datasets.

**Weaknesses:**

- While this patch-level performance is impressive, the majority of computational pathology tasks are performed at the slide level. Slide-level performance using ABMIL and mean pooling should be shown. In addition to C16 and TCGA-LUSC, which have corresponding slide labels from more WSIs than used for the patch-level tasks, the PANDA dataset can be utilized, which can be divided between the Radboud and Karolinska institutes. If used, please report quadratic weighted kappa.

- While the empirical results indicate that the patch embeddings based on GLMP are relatively agnostic to tissue site, it remains unclear why the text captions would not be subject to those tissue site biases. For instance, could it be that variations in brightness, staining color, etc, affect the quality of the captions, and subsequently, the model’s ability to detect subtle changes such as mitotic figures?

- Figure 2 is important for building intuition on tissue site biases. Please show examples on an additional dataset in the appendix.

- ***The examined tasks all relate to tissue morphology, which have clearly-defined criteria for each tissue class. Meanwhile, I suspect the model will struggle on tasks such as molecular subtyping or survival prediction, which do not have as clearly-defined criteria and will require more detailed textual descriptions. Ideally, multiple target tasks on a single WSI dataset (e.g TCGA NSCLC prediction of tp53, kras, egfr) would substantially improve the reliability of these results, showing that a single set of captions can encode sufficient information for a wide range of tasks. I believe this is the most important component missing from the paper.

- Please define the meaning of “cross-TSI generalization” in Line 259. While GLMP exhibits a smaller drop in performance from training to test site, Figure 4 shows that GLMP achieves lower overall performance on both the internal and external institution.

- Line 269 is not sufficiently supported in my opinion. There are a large number of differences between c16 and TCGA-LUSC, and so the conclusion that GLMP performs better on more challenging tasks requires additional experiments, such as evaluation on molecular subtyping tasks.

- Clarity can be substantially improved by describing the procedure for creating tissue site correlations  (Lines 299-301) within the paragraph it is introduced.

- The results in Figure 5 are misleading by including 100/0 high bias. While the 100/0 bias case does indicate GLMP’s efficacy at removing tissue site bias, this tissue site invariance is already extensively shown in section 4.6. Meanwhile, differences at lower bias ranges are of substantially higher interest to the research and clinical community. Inclusion of this 100/0 case disrupts the scale, making differences at more realistic bias rations difficult to discern. Please move the 100/0 bias case from the main figure into a supplemental figure, and include additional experiments in the 50-90 range.

- Performance on Tissue4Skin and TumSeg should be included in Figure 4.

- Table 4 is an important result. This ablation should be performed on additional datasets.

**Questions:**

- Why does institution 3’s image look strange? There appears to be some filter applied on top of the image.

- What happens to performance when you exclude the explicit instruction to include non-biological characteristics (Lines 426-428).

# Additional Feedback
- In the introduction, it would be more clear to distinguish between patch foundation models and slide foundation models.

- Figure 1 could be aesthetically improved

- In section 4.2, please specify that k-means is performed inter-slide.

- The captions should be more detailed. For instance, figure 2 should specify the tissue type and dataset, and that k-means is performed inter-slide.

---

### Official Review · Reviewer_WoM7 · 2025-10-30

**Soundness:** 2
**Presentation:** 3
**Contribution:** 2
**Rating:** 4
**Confidence:** 4

**Summary:**

This paper introduces GLMP, a model that tackles non-biological "batch effects" (like stain variations) that cause pathology models to fail across different institutions. Instead of encoding images directly, GLMP uses a multimodal LLM to generate a text description focused only on biological features, filtering out the batch effect artifacts. This text-based embedding proves more robust and generalizable than standard models.

**Strengths:**

1.	This work proposes new inspiration for mitigating batch effects inside pretrained pathology foundation models.

2.	Experimental analysis is comprehensive, and the proposed solution seems effective.

**Weaknesses:**

1.	The logic of this paper is that, to mitigate batch effects (assumed), the pretrained pathology foundation models are completely discarded, and instead, a general-purpose MLLM such as Gemini-2.5-Pro is used to first describe the pathology image and then generate the feature of it, with no measure to ensure the correctness of the description. This seems to neglect the essential and pursue the trivial.

2.	The efficiency of the proposed workflow is a major concern. Describing every patch in a WSI with Gemini-2.5-Pro and then projecting it into a feature vector should take much more time compared to pre-trained pathology foundation models. And I suggest the authors add an efficiency comparison between the proposed model and baselines.

3.	I acknowledge the importance of the proposed research question; however, whether the proposed pipeline is the best way to solve it remains a question. The proposed method seems too preliminary, and it cannot be deployed to scenarios like WSI-level diagnosis or prognosis. Moreover, the contribution came more from Gemini's powerful capabilities. I suggest the authors read this work [1], which also works on the robustness of pathology foundation models, to understand how to come up with their own contribution.

$\quad$ [1] Kömen, Jonah, et al. "Towards Robust Foundation Models for Digital Pathology." arXiv preprint arXiv:2507.17845 (2025).

**Questions:**

1.  Could the authors elaborate more on the choice of leveraging general-purpose MLLMs instead of pathology-specific MLLMs [1,2]?

$\quad$ [1] Xu, Zhe, et al. "A versatile pathology co-pilot via reasoning enhanced multimodal large language model." arXiv preprint arXiv:2507.17303 (2025).

$\quad$ [2] Sun, Yuxuan, et al. "Cpath-omni: A unified multimodal foundation model for patch and whole slide image analysis in computational pathology." Proceedings of the Computer Vision and Pattern Recognition Conference. 2025.

2.	The authors mentioned “A structured prompt guides the MLLM to describe only biological characteristics, such as cellular morphology and tissue architecture,” in the last paragraph of Introduction. But how to ensure the accuracy of the statement generated by MLLMs?

3.	In the Cross-TSI Generalization task, why choose linear probe rather than MIL models such as ABMIL for evaluation?

4.	The authors mentioned that “However, reliance on large-scale annotated pathology data makes these methods non-scalable given the scarcity and cost of such data.” in the first paragraph of Related Work. However, the proposed method leverages the MLLMs to describe the pathology images. In this sense, researchers could also use MLLMs to caption unlabeled pathology images to create scalable image-text paired datasets. The logic here does not really work out.

5.	Could the author please add these papers to the experimental comparison since they are relevant:

$\quad$ [1] Wang, Xiyue, et al. "A pathology foundation model for cancer diagnosis and prognosis prediction." Nature 634.8035 (2024): 970-978.

$\quad$ [2] Ma, Jiabo, et al. "A generalizable pathology foundation model using a unified knowledge distillation pretraining framework." Nature Biomedical Engineering (2025): 1-20.

$\quad$ [3] Xu, Yingxue, et al. "A multimodal knowledge-enhanced whole-slide pathology foundation model." arXiv preprint arXiv:2407.15362 (2024).

---

### Official Review · Reviewer_a5So · 2025-10-31

**Soundness:** 2
**Presentation:** 3
**Contribution:** 2
**Rating:** 2
**Confidence:** 4

**Summary:**

The submission proposes a method for encoding histopathology image patches into a feature representation that is invariant to batch effects. Batch effects in medical datasets arise when datasource signatures are present in data collected in separate batches, potentially confounding predictions.

Overall, the method involves asking a multimodal LLM to describe an input pathology image in text, and then encoding the text with a text encoder. This encoding of the generated text is then used to showcase better independence of information from tissue-source-site through separability-analysis with visualization tools such as clustering/PCA on the embeddings. Experiments with tissue-classification further confirm that the proposed method is less confounded by tissue source information, relative to pathology foundation models that provide image embeddings.

**Strengths:**

The submission explores a very critical problem when working in data-variability-constrained areas in medical AI such as histopathology, namely batch effects.

The key takeaway for me is that modern day LLMs might already be useful at extracting clinically useful information without being overwhelmed by extraneous information.

**Weaknesses:**

The fundamental weakness in this work is that it does not adequately explore the way by which the MLLM, Gemini in this case, actually enables agnosticism to batch effects to how generalizable the overall approach is likely to be.

To my knowledge, we do not know what data-sources are actually used to train Gemini, and how. The submission posits it is "general image-text data", but given the clinically-oriented outputs, it seems likely that the training involved a fair bit of histopathology data, perhaps even including all of the data used for the experiments in the submission. The prompts are very specifically asking for clinically-oriented outputs, thus it seems quite likely that the information being assessed is already within Gemini training, and already perfectly-aligned with the tests in the experiments. Therefore, the submission may only be demonstrating that Gemini is good at histopathology, and it is unclear if the overall methodology for avoiding batch-effects is sound, i.e. what if one trained an MLLM from scratch on limited data carefully curated to exhibit batch-effects from train to test sets? The lack of visibility into the chosen MLLM makes the results hard to interpret in terms of overall methodology.

**Questions:**

None

---

### Note · Authors · 2025-11-14

**Comment:**

The authors thank the reviewers for their helpful feedback.

**Withdrawal Confirmation:**

I have read and agree with the venue's withdrawal policy on behalf of myself and my co-authors.